# Development and Validation of a Highly Sensitive LC-MS/MS Method for the Analysis of Bile Acids in Serum, Plasma, and Liver Tissue Samples

**DOI:** 10.3390/metabo10070282

**Published:** 2020-07-09

**Authors:** Cristina Gómez, Simon Stücheli, Denise V. Kratschmar, Jamal Bouitbir, Alex Odermatt

**Affiliations:** Division of Molecular and Systems Toxicology, Department of Pharmaceutical Sciences, University of Basel, Klingelbergstrasse 50, 4056 Basel, Switzerland; cristina.gomezcastella@unibas.ch (C.G.); simon.stuecheli@unibas.ch (S.S.); denise.kratschmar@unibas.ch (D.V.K.); jamal.bouitbir@unibas.ch (J.B.)

**Keywords:** bile acid, LC-MS, statin myotoxicity, biomarker, adverse drug effect, disease

## Abstract

Bile acids control lipid homeostasis by regulating uptake from food and excretion. Additionally, bile acids are bioactive molecules acting through receptors and modulating various physiological processes. Impaired bile acid homeostasis is associated with several diseases and drug-induced liver injury. Individual bile acids may serve as disease and drug toxicity biomarkers, with a great demand for improved bile acid quantification methods. We developed, optimized, and validated an LC-MS/MS method for quantification of 36 bile acids in serum, plasma, and liver tissue samples. The simultaneous quantification of important free and taurine- and glycine-conjugated bile acids of human and rodent species has been achieved using a simple workflow. The method was applied to a mouse model of statin-induced myotoxicity to assess a possible role of bile acids. Treatment of mice for three weeks with 5, 10, and 25 mg/kg/d simvastatin, causing adverse skeletal muscle effects, did not alter plasma and liver tissue bile acid profiles, indicating that bile acids are not involved in statin-induced myotoxicity. In conclusion, the established LC-MS/MS method enables uncomplicated sample preparation and quantification of key bile acids in serum, plasma, and liver tissue of human and rodent species to facilitate future studies of disease mechanisms and drug-induced liver injury.

## 1. Introduction

Bile acids are important for the digestion and absorption of lipids and fat-soluble vitamins in the intestine but also for the regulation of cholesterol homeostasis in the liver. Besides, they are bioactive molecules signaling through receptors such as farnesoid X receptor, pregnane X receptor, vitamin D receptor, sphingosine-1-phosphate receptor 2, and G protein-coupled bile acid receptor 1 in order to modulate lipid and glucose homeostasis, with relevance to cardio-metabolic disease and cancer [1,2]. Several individual bile acids are considered biomarkers of disease or drug-induced liver injury (DILI) [3,4,5] and improved quantification methods are therefore needed.

Bile acids are steroidal C24 carboxylic acids, synthesized in the liver from cholesterol. In humans, the two primary bile acids cholic acid (CA) and chenodeoxycholic acid (CDCA) are synthesized in hepatocytes by the classical and alternative pathway, respectively [6,7] (Figure 1). In rats and mice, the primary bile acids consist mainly of CA and the muricholic acid (MCA) metabolites αMCA and βMCA that are derived by cytochrome P450 (CYP) 2C70 from CDCA and ursodeoxycholic acid (UDCA), respectively [8]. In guinea pigs, 7-oxolithocholic acid (7oxoLCA) is a major primary bile acid, in contrast to human, rat, and mouse where it is a secondary bile acid and only found at low levels under healthy conditions [9]. The newly synthesized bile acids are conjugated on the side chain to mainly glycine in humans and taurine in rodents, and to a lesser extent on the steroid nucleus to sulfate and glucuronide, followed by excretion mainly through the bile [6,7]. In the intestine, the primary bile acids undergo modifications by intestinal bacteria through deconjugation, dihydroxylation, epimerization, and oxidation, resulting in secondary bile acids including lithocholic acid (LCA), deoxycholic acid (DCA), and their 7-oxo forms, as well as UDCA [10,11]. Secondary bile acids also undergo conjugation to taurine and glycine in the liver and intestine. Due to the function of the steroid 5β-reductase (aldo–keto reductase (AKR)1D1) all major bile acids are 5β-reduced. However, some bile acids escape AKR1D1 metabolism, leading to allo-bile acids with a α- rather than β-oriented hydrogen at C5 [10,12].

An efficient bile acid transport by hepatocytes and terminal ileal enterocytes mediates recycling of bile acids by the enterohepatic circulation [13,14]. This is a highly efficient process and only about 5% of total bile acids are lost via excretion through the feces. The excreted amount is replenished by de novo synthesis to preserve a constant bile acid pool [15]. Disruption of bile acid transport, for example by inhibition of hepatic mitochondrial function of bile acid export proteins such as a bile salt export pump (BSEP), can cause cholestasis with accumulation of bile acids in hepatocytes and then provoke liver injury due to their detergent-like effects [16,17]. Cholestasis also results in increased circulating bile acid levels with distinct alterations in individual bile acid metabolites [2,5,18,19].

An increase in circulating total bile acids is an accepted biomarker of hepatobiliary impairment and disease [17,20]. However, the hydrophobic LCA and DCA are considered the most cytotoxic bile acids [17], and their accumulation causes impairments in hepatocytes such as mitochondrial dysfunction, oxidative stress, endoplasmic reticulum stress, and damage and disruption of cell membranes leading to apoptosis and necrosis [20,21]. A recent study observed that the ratio of taurine- and glycine-conjugated CA and CDCA over their free forms were better predictors of DILI due to inhibited mitochondrial function and BSEP transport than total bile acids [22]. Thus, the measurement of individual bile acids can provide important mechanistic understanding. Bile acids can also serve as functional biomarkers providing information on deficiencies of specific enzymes involved in the metabolism of bile acids [23,24,25,26]. The development and validation of methods for targeted quantification of bile acids are essential to understand their mechanisms of action in vivo. Moreover, the diagnosis of rare diseases related to cholesterol biosynthesis and metabolism as well as for the detection of DILI can be accomplished by the determination of bile acid profiles in body fluids.

Over the last decade, several methods using different platforms, including liquid chromatography-tandem mass spectrometry (LC-MS/MS) based methods, have been reported for bile acid separation, detection, and quantification [27,28,29,30,31,32,33,34,35,36]. The simultaneous analysis of bile acids in biological samples such as serum, plasma, and liver biopsies is challenging because of their structural similarities and limits of detection. Most of the developed methods exert some disadvantages, mainly caused by a series of laborious sample preparation steps, limited sensitivity of detection, or a lack of specificity due to insufficient chromatographic baseline separation of isobaric bile acids. Accordingly, highly sensitive and specific methods to quantify bile acids in biological samples are needed.

The present study aimed to develop and validate a highly sensitive and specific method for the rapid and simultaneous quantification of a number of bile acids by LC-MS/MS. Compared to previously reported methods, the first goal was to achieve an improvement of the method in terms of simplicity, metabolic coverage by including free and conjugated metabolites, and applicability for different matrices and species. A second goal included the validation of the method using real samples. For this purpose, the developed method was applied to quantify bile acid profiles in plasma and liver tissue samples from mice treated for three weeks with increasing doses of the 3-hydroxy-3-methylglutaryl coenzyme A (HMG-CoA) reductase inhibitor simvastatin. Although statins are in general well tolerated in patients, they can lead mainly to skeletal muscle side effects. The clinical manifestations start from myalgia, myopathy to potentially fatal rhabdomyolysis in rare cases [37,38]. These adverse skeletal muscle effects were observed at the doses used in this study [39,40,41]. Therefore, we assessed whether inhibition of *de novo* cholesterol formation [38,42] and the production of intermediates of cholesterol synthesis such as ubiquinone (CoQ10) and dolichols [38] by simvastatin might disrupt bile acid homeostasis and contribute to myotoxicity.

## 2. Results and Discussion

### 2.1. Establishment of the LC-MS/MS Method for Quantification of Bile Acids

The present study employed LC-MS/MS for the direct detection and quantification of 36 different free, taurine- and glycine-conjugated bile acids in serum, plasma, and liver tissue. These analytes include important bile acids of human (free and mainly glycine-conjugated) and rodents (free and mainly taurine-conjugated) but also some additional bile acids found in guinea-pig (7oxoLCA and its conjugated forms) and such being relevant with regard to activities of specific enzymes (7oxoLCA, allo-bile acids, and their conjugated forms) [9,10,11,12]. In order to select the best transitions for the identification and confirmation of each bile acid included in the method, individual working standard solutions were applied to the optimizer compound software. The best collision energy was obtained for each multiple reaction monitoring (MRM) transition. Both positive and negative ionization modes were carried out. For each bile acid, the transition with the highest sensitivity was chosen as a quantifier (Table 1) and at least one other transition was selected as a qualifier for each compound. LC gradient elution pattern was optimized in order to achieve baseline separation for all compounds. Different column temperatures and flow rates were tested and the parameters resulting in better separation were selected (see section LC-MS/MS instrumental conditions). Retention times are listed in Table 1. MS parameters such as desolvation gas flow, sheath gas flow, nebulizer pressure, capillary voltage, and nozzle voltage were chosen using the optimizer software of the instrument. Optimal parameters are described in the materials and methods section. 

The current method contains multiple improvements compared to previously published methods [27,28,29,30,31,32,33,34,35]. One of the main advantages is the incremented metabolic coverage including free and conjugated bile acids, as well as allo-bile acids and 7-oxo metabolites [27,28,29,31,32,34,35]. A better baseline chromatographic separation of the isobaric metabolites was achieved [23,29,30,31,33,35]. Furthermore, lower values for limit of detection (LOD) and limit of quantification (LOQ) were observed for most of the metabolites analyzed in the present study (Table 1 and Table 2) [28,31,34,35], despite using a lower injection volume into the LC-MS/MS system [27,28,30,31,33,34,35]. The amount of sample material required was lower compared to similar reported methods [27,28,30,31,32,33,34,35], which represents a great advantage for studies using rodent models where usually limited sample volume is available.

### 2.2. Method Validation

The method was then validated following the recommendations for bioanalytical method validation [43] (Table 1 and Table 2) and confirmed to be selective and specific. Specifically, the absence of interfering substances at the retention times of the compounds of interest and internal standards were verified in charcoal-stripped plasma and charcoal-stripped serum.

LOD and LOQ were determined for all compounds for both extraction protocols. Using the extraction protocol for plasma and serum samples, the LOD ranged from 0.01 to 1 ng/mL and the LOQ ranged from 0.02 to 3.5 ng/mL (Table 1). The LOD for the liver extraction protocol ranged from 0.03 to 7 ng/mL and the LOQ from 0.09 to 21 ng/mL (Table 2). Carryover was assessed by injecting five consecutive blank samples after a calibration standard at the upper limit of quantification, for both extraction protocols. No carryover could be detected for the analytes and the internal standards, as there were no chromatographic peaks present at the transitions at the corresponding retention times. Linearity of the calibration curves was assessed by using linear regression analysis of a triplicate of the 10-level calibration curve prepared independently for both extraction protocols. All calibration curves were linear, with squared correlation coefficients (r2) ranging from 0.990 – 0.999 for the majority of the compounds in both extraction protocols, except for allo-3β-lithocholic acid (allo-3βLCA) using the serum/plasma extraction protocol and LCA and tauro-lithocholic acid (TLCA) with the liver extraction protocol (Table 1 and Table 2). Extraction recovery was evaluated by comparing the responses of a spiked matrix with and without extraction, and calculated for each analyte at low, medium, and high concentrations (2.5, 50, and 250 ng/mL) (Table 1 and Table 2). Stability was determined by analyzing extracted samples and calibrators immediately as well as after storage at 4 °C for one week. Concentrations obtained were compared and variations lower than 10% were obtained throughout, regardless of the storage period. All bile acids were found to be stable at 4 °C for at least one week.

The method was then tested using different matrices from human and mouse (Figure 2). The list of bile acids included in the LC-MS/MS method for quantification is represented by three different colors, indicating the robust detection and quantification of the compound (green), presence at a low concentration around LOQ, or absence in few tested samples (yellow), and absence in most samples (red) (concentrations at Appendix A). In contrast to humans, where glycine-conjugated bile acids are the major components of the bile acid pool with a ratio of glycine to taurine-conjugated bile acids of about 3:1 in adult males, taurine conjugates represent the most abundant bile acids in the mouse and rat [7]. All primary and secondary bile acids could be robustly quantified, with the exception of LCA and its conjugates. Some of the allo-bile acids and 7oxo bile acids were either present at low concentrations around LOQ or not detected. This is not unexpected, as these bile acids represent minor metabolites that are thought to be increased in certain enzyme deficiencies or diseases [23,44].

### 2.3. Application of the Method

The method was applied to plasma and liver tissue samples from mice treated for three weeks with simvastatin at three different doses (5, 10, and 25 mg/kg/d). Simvastatin, similar to other statins, inhibits HMG-CoA reductase that catalyzes the rate-limiting step of cholesterol biosynthesis and is widely used to treat hypercholesterolemia [45]. The main adverse effects of statins are skeletal muscle-associated symptoms, seen in up to 30% of treated patients, with rhabdomyolysis as a rare but potentially fatal form [38]. Since bile acids are produced from cholesterol, we tested the hypothesis that inhibition of hepatic cholesterol synthesis by simvastatin might alter bile acid homeostasis and that this potentially contributes to statin-induced myotoxicity. In mice treated daily for three weeks at 5 mg/kg, grip strength and muscle endurance capacity were decreased while plasma lactate levels were elevated after exercise compared to control mice. Additionally, disturbances of mitochondrial function could be detected with impaired oxidative metabolism [39,40].

More than 20 different bile acids were detected and quantified in plasma and liver tissue of simvastatin treated mice, including free (Figure 3A,C) and taurine-conjugated (Figure 3B,D) metabolites (concentrations at Appendix A). As expected, taurine-conjugated bile acids were predominant in mouse, whereas glycine-conjugated metabolites were either undetectable or present at low levels [6,7] and therefore not further considered. The results showed no significant changes between simvastatin treated and control mice in the concentrations of any bile acid, including major and minor metabolites. To better understand our observation, we measured plasma lipid profile including total cholesterol, high-density lipoprotein cholesterol (HDLc), and low- and very low-density lipoprotein cholesterol (LDLc/VLDLc) (Appendix A). We did not detect any significant difference between simvastatin-treated and control mice in the aforementioned lipid markers. These results are in accordance with a previous study in which the treatment with simvastatin at 30 mg/kg for two weeks did not change the lipid profile in mice [46]. Since bile acids are amphipathic molecules derived from cholesterol, the unchanged lipid profile could be the obvious reason why the treatment with simvastatin at the tested doses did not change the bile acid profile in mice. Moreover, these results are in line with an earlier study of Fu et al., in which they treated mice with atorvastatin at 100 mg/kg/d for one week [47]. Although they observed an induction of Cyp7a1, the rate-limiting enzyme in bile acid synthesis, total bile acids in serum and liver tissue did not change. However, their data did not exclude the possibility that individual bioactive bile acids might be altered. In the present study, application of the newly established bile acid quantification method, covering a large number of metabolites, did not show any significant differences in the free and taurine-conjugated bile acids measured in both plasma and liver tissue (Figure 3). This data is in line with observations in other studies using different species such as rabbit and human patients, where simvastatin did not alter the bile acid profiles [48,49]. Together, the data indicate that the cholesterol-lowering effect of simvastatin does not result in disruption of bile acid homeostasis and therefore is not involved in the observed simvastatin-induced myotoxicity.

## 3. Materials and Methods 

### 3.1. Chemicals and Reagents

Ultrapure water was obtained using a Milli-Q^®^ Integral 3 purification system equipped with an EDS-Pak^®^ Endfilter for the removal of endocrine active substances (Merck Millipore, Burlington, MA, USA). Acetonitrile (HPLC-S Grade) was purchased from Biosolve (Dieuze, France), chloroform stabilized with ethanol (HPLC Grade) from Scharlab (Sentmenat, Spain), methanol (CHROMASOLV™ LC-MS grade) from Honeywell (Charlotte, NC, USA), isopropanol (EMSURE^®^ for analysis) from Merck Millipore and formic acid (Puriss. p.a. ≥ 98%) from Sigma-Aldrich (St. Louis, MO, USA).

7oxoLCA was purchased from BioTrend (Köln, Deutschland), tauro-cholic acid, tauro-ursodeoxycholic acid, and glyco-cholic acid were obtained from Calbiochem (San Diego, CA, USA), and tauro-chenodeoxycholic acid, CA, UDCA, glyco-chenodeoxycholic acid, glyco-deoxycholic acid, DCA, CDCA, and LCA from Sigma-Aldrich. Tauro-α-muricholic acid, tauro-β-muricholic acid, tauro-ω-muricholic acid, αMCA, βMCA, ωMCA, 7-oxodeoxycholic acid, glyco-ursodeoxycholic acid, tauro-deoxycholic acid, hyodeoxycholic acid, TLCA, glyco-lithocholic acid, dehydrolithocholic acid, 12-oxolithocholic acid, 6,7-dioxolithocholic acid, and γMCA were obtained from Steraloids (Newport, RI, USA). Tauro-7-oxolithocholic acid and glyco-7-oxolithocholic acid. Allo-cholic acid, allo-deoxycholic acid, allo-lithocholic acid, allo-3β-deoxycholic acid, allo-3β-LCA, and allo-12β-deoxycholic acid were kindly provided by Prof. Alan F. Hofmann (San Diego). Deuterated standards used as internal standards, i.e., (2,2,4,4-2H4)-glyco-chenodeoxycholic acid, (2,2,4,4-2H4)-glyco-cholic acid, (2,2,4,4-2H4)-glyco-ursodeoxycholic acid, and (2,2,4,4-2H4)-UDCA were obtained from CDN Isotopes (Pointe-Claire, Quebec, Canada). (2,2,4,4-2H4)-CA, (2,2,4,4-2H4)-CDCA, and (2,2,4,4-2H4)-LCA were purchased from Sigma-Aldrich. (2,2,4,4-2H4)-DCA was purchased from Steraloids.

### 3.2. Serum, Plasma, and Liver Tissue Samples from Mice and Plasma from Humans Used for Method Development

Blood and liver tissue samples of 10–12 weeks old male C57BL/6J mice were collected in agreement with the guidelines for care and use of laboratory animals, accepted by the cantonal veterinary authority of Basel in Switzerland (License 2758).

Human plasma from 16 healthy subjects (8 men and 8 women; mean ± SD age 24.8 ± 2.6 years) was obtained from a previously reported study [50] conducted in accordance with the Declaration of Helsinki and International Conference on Harmonization Guidelines in Good Clinical Practice and approved by the local Ethics Committee and Swiss Agency for Therapeutic Products (Swissmedic). The study was registered at ClinicalTrials.gov: http://clinicaltrials.gov/ct2/show/NCT01465685.

### 3.3. Mice Treated with Simvastatin

Male C57BL/6J mice (*n* = 24, 14 weeks old, approximately 29–32 g of bodyweight) were acclimatized one week prior to the start of the study and housed in a standard facility with 12 h light-dark cycles and controlled temperature (22 ± 2 °C). The mice were fed a standard pellet chow and water ad libitum. All experiments were performed in agreement with the guidelines from Directive 2012/63/EU of the European Parliament on the protection of animals used for scientific purposes. The experiments performed in this animal study were reviewed and accepted by the cantonal veterinary authority of Basel in Switzerland and were performed in agreement with the guidelines for care and use of laboratory animals (License 3035).

After acclimatization, mice were randomly divided into four groups: (1) mice treated with vehicle (control, *n* = 6); (2) mice treated with simvastatin at 5 mg/kg/d (*n* = 6); (3) mice treated with simvastatin at 10 mg/kg/d (*n* = 6); and (4) mice treated with simvastatin at 25 mg/kg/d (*n* = 6). The mice were treated by oral gavage for three weeks. Following treatment for 21 days, mice were anesthetized with an intraperitoneal injection of ketamine (100 mg/kg) and xylazine (10 mg/kg) (Graeub AG, Switzerland). Then, blood was collected into heparin-coated tubes by intracardiac puncture. Plasma was separated by centrifugation at 3000× *g* for 15 min. Plasma samples were kept at −80 °C for later analysis. The liver was immediately collected, frozen in liquid nitrogen, and stored at −80 °C for later analysis.

### 3.4. Bile Acids Extraction from Plasma and Serum

For the analysis of bile acids, 25 µL of plasma and serum samples diluted at a ratio of 1:4 (*v/v*) with MilliQ water were used. Samples were subjected to protein precipitation by adding 1 mL of 2-propanol containing a mixture of deuterated internal standards. Extraction was performed with continuous shaking at 4 °C for 30 min at 1400 rpm on a Thermomixer C (Eppendorf AG, Hamburg, Germany) and then centrifuged at 16000× *g* for 10 min. Supernatants were transferred to new tubes and evaporated to dryness by using a Genevac EZ-2 system (SP Scientific, Warminster, PA, USA) at 35 °C. Extracts were resuspended in 100 µL methanol:water (1:1, *v*/*v*), incubated at 4 °C for 10 min at 1400 rpm, sonicated in a water bath for 10 min at room temperature, and finally transferred to LC-MS vial for analysis.

### 3.5. Bile Acids Extraction from Mouse Liver Tissue

Liver samples (30 ± 5 mg) were homogenized using oxide beads (1.4 mm Zirconium) on the Precellys 24 Tissue Homogenizer (Bertin Instruments, Montigny-le-Bretonneux, France) with three cycles (30 s at 6500 rpm, 30 s break between each cycle) in 1 mL of extraction mixture (water-chloroform-methanol; 1:1:3; *v*/*v*/*v*) including deuterated internal standards. Samples were incubated for 15 min at 37 °C with continuous shaking at 850 rpm on a Thermomixer C and centrifuged at 16,000× *g* for 10 min at 20 °C. 800 µL of supernatant were transferred to a 2 mL tube and kept on ice. Another 800 µL of extraction mixture were added to the samples and the extraction process was repeated once more as described. The final 1.6 mL were evaporated to dryness in a Genevac EZ-2 at 35 °C. The residue was reconstituted in 200 µL methanol-water 1:1 (*v*/*v*), incubated for 10 min at 1300 rpm and at 20 °C, and then sonicated in a water bath for 10 min at room temperature. Next, the samples were centrifuged at 16,000× *g* for 10 min at 20 °C and 100 µL of the supernatant were transferred to LC-MS vials and stored at −20 °C for later analysis.

### 3.6. Calibration and Quality Control Standard Preparation

Bile acid standard or internal standard stock solutions at concentrations of 1 mg/mL were prepared by dissolving each standard in methanol and stored at −80 °C. Individual working solutions were prepared by diluting them with the appropriate amount of methanol. A mixture of all internal standards was prepared for sample preparation. A mixture of all the compounds was prepared to spike the highest level of the calibration curve. A ten-point calibration curve was prepared by serial dilution of the working standard mix solution into charcoal-stripped serum, plasma, or phosphate-buffered saline (PBS). Quality controls were prepared by spiking the working standard solution mix (at 2.5, 50, and 100 ng/mL) in charcoal stripped serum, plasma or PBS. Calibrators and quality controls then went through the sample preparation process described above.

### 3.7. LC-MS/MS Instrumental Conditions

The analysis was carried out using an Agilent 1290 UPLC coupled to an Agilent 6490 triple quadrupole mass spectrometer equipped with an electrospray ionization (ESI) source (Agilent Technologies, Basel, Switzerland). Drying gas as well as nebulizing gas was nitrogen. The desolvation gas flow was set at 15 L/min, the sheath gas flow at 11 L/min, and the nebulizer pressure at 20 psi. The nitrogen desolvation temperature was set at 290 °C, sheath gas temperature at 250 °C. Capillary voltage was optimized for each segment from 2000 to 5000 V. Nozzle voltage was set at 2000 V and cell accelerator voltage at 5 V. Chromatographic separation of bile acids was achieved using reversed-phase column (ACQUITY UPLC BEH C18, 130 Å, 1.7 µm, 2.1 mm × 15 mm, Waters, Milford, MA, USA), at a flow of 0.5 mL/min and a column temperature of 55 °C. The mobile phase consisted of ultrapure water and acetonitrile (95:5, *v*/*v*) with 0.1% formic acid (solvent A) and acetonitrile and ultrapure water (95:5, *v*/*v*) with 0.1% formic acid (solvent B). The following gradient pattern was used for the separation of bile acids: 0 min, 25% B, 3.1 min, 35% B, 9 min, 38% B, 15 min, 65% B, 18 min, 65% B, 20 min, 100% B, 22 min, 25% B, and additional 2 min post-run at initial conditions. The injection volume was set at 3 µL.

Data acquisition was performed using MRM mode. At least two transitions (quantifier and qualifier transitions) were selected for each compound in positive or negative ESI mode depending on the compound. Collision energy was optimized for each transition (Table 1 and Table 2).

### 3.8. Method Validation

The following parameters were evaluated: LOD, LOQ, linearity, stability at 4 °C, carryover, and extraction recovery. The absence of any interfering substance at the retention time of the compounds of interest was verified [43].

The LOQ was defined as the lowest concentration at which the peak response was ten times that of the noise (10 S/N), and the LOD was the extrapolated concentration with a signal-to-noise ratio of three (3 S/N). Calibration curves were prepared daily and spiked with internal standards. In order to evaluate the linearity, 10 calibration curve points were analyzed by the optimized method. The linearity of the calibration curves was then determined using linear regression analysis of a triplicate of the 10-level calibration curve prepared independently for both extraction protocols described above. Carryover was assessed by injecting five consecutive blank samples after injection of the highest calibration standard for both extraction protocols. The stability of the bile acids at 4 °C was evaluated in plasma and liver tissue samples to investigate whether the concentrations of the compounds were stable over time. Extracts were analyzed, kept at 4 °C for one week, and reanalyzed.

The extraction recovery was assessed at three concentration levels (2.5, 50, and 250 ng/mL). The analysis of six replicates of a charcoal-treated plasma sample spiked with the compounds before extraction, and six replicates of the same charcoal treated plasma sample to which the analytes were added after extraction. The ratio of the peak areas between the analytes and the internal standard obtained from the extracted spiked samples was compared with ratios obtained for samples in which the analytes were added after extraction of the matrix (representing 100% extraction recovery). In order to evaluate the extraction recovery for the liver sample extraction protocol bovine serum albumin (1 mg/mL) was used as a matrix. Precision was measured as the relative standard deviation of the ratios of the peak areas of the compound to the internal standard. The concentration range for the quantification was optimized for each matrix.

The extraction protocols were applied to different matrices in order to evaluate its performance before being applied to a real cohort of samples. The method was tested in human and mouse serum and plasma as well as mouse liver tissue samples.

### 3.9. Data Analysis and Statistics

For LC-MS/MS data, MassHunter Acquisition and Quantitative Analysis vB.07.01 (Agilent Technologies, Inc.) were used for quantification. Statistical analysis was performed using nonparametric multiple *t*-tests, groups were compared by using the Wilcoxon signed rank test. Statistical analysis and graphs were performed in GraphPad Prism v5.02 (GraphPad Software).

## 4. Conclusions

We developed and validated an LC-MS/MS method for the simultaneous quantification of 36 different bile acids in serum, plasma, and liver tissue samples. This simple method solved issues of separation of isobaric bile acids seen in previous methods and its applicability was demonstrated using human and mouse samples. The method covers the most important free and taurine- and glycine-conjugated bile acids as well as some minor metabolites of interest in certain enzyme deficiencies. This broad coverage and the small sample amount needed allows future application of the method for clinical studies as well as preclinical studies including species such as mouse, rat, and guinea-pig. Finally, application of the method in a study of simvastatin treated mice showed that the cholesterol-lowering drug, at doses causing myotoxicity, did not affect bile acid profiles in plasma and liver tissue, indicating that bile acids are not involved in the mechanism of statin-induced myotoxicity. 

## Figures and Tables

**Figure 1 metabolites-10-00282-f001:**
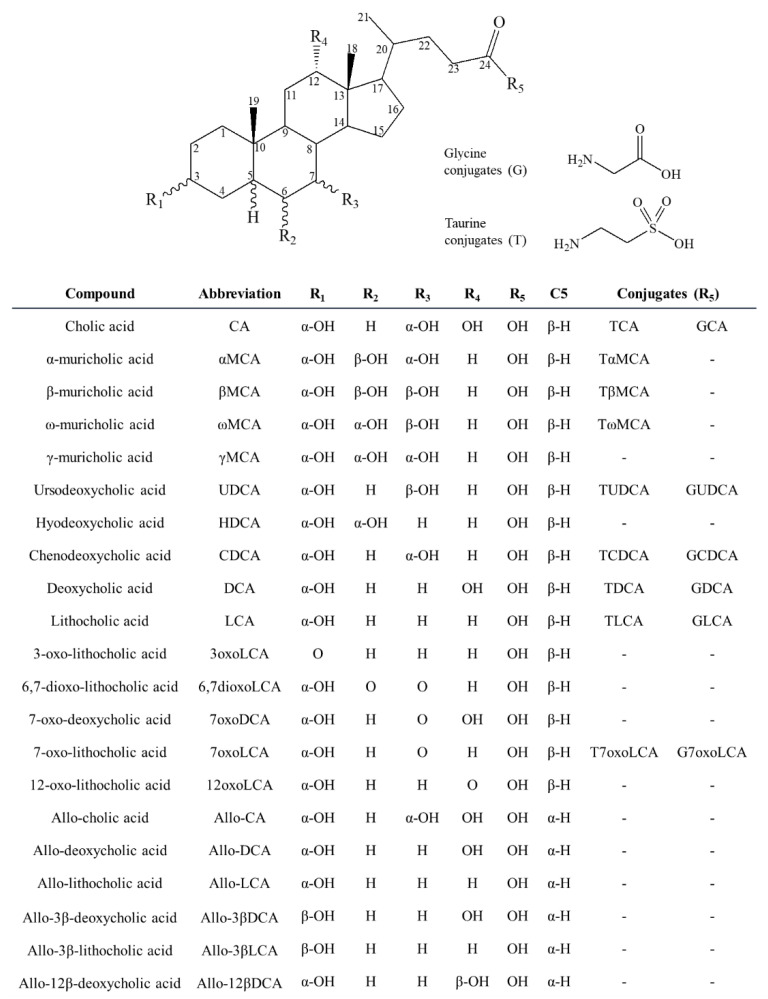
Chemical structures of bile acids included in the LC-MS/MS method.

**Figure 2 metabolites-10-00282-f002:**
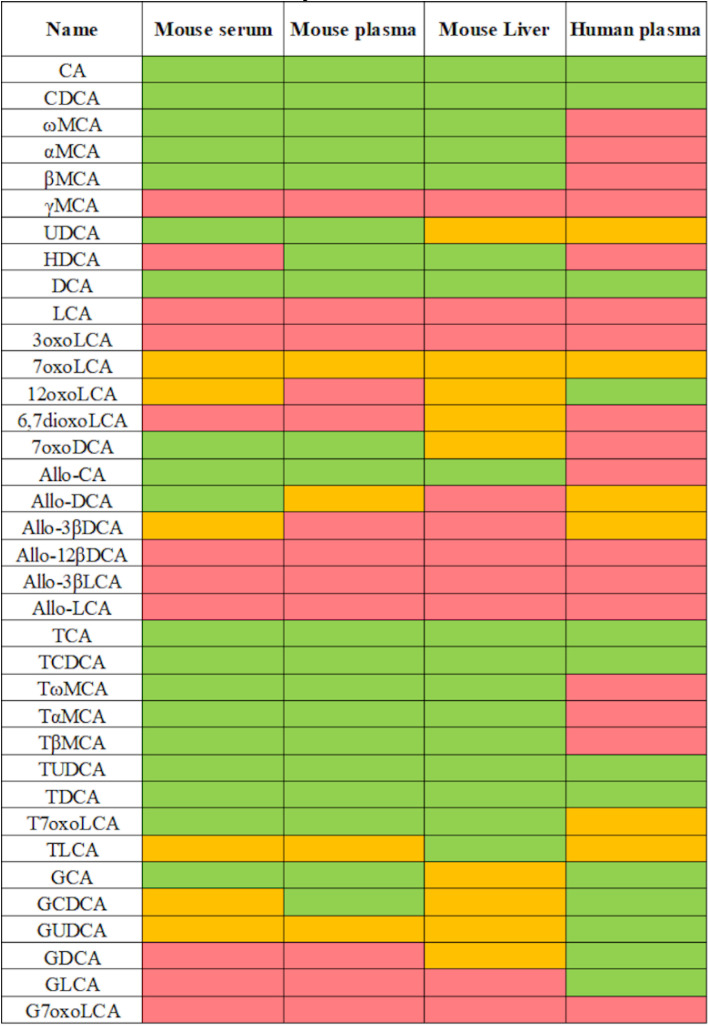
List of bile acids included in the LC-MS/MS method for quantification. The newly established method was applied to different matrices including plasma of human and mouse, as well as mouse serum and liver tissue. Colors indicate compound detection and quantification: Green: Compound detected in samples used for this test. Yellow: Compound detected at a very low concentration, close to LOQ or not detected in all samples tested. Red: Compound not detected in most samples.

**Figure 3 metabolites-10-00282-f003:**
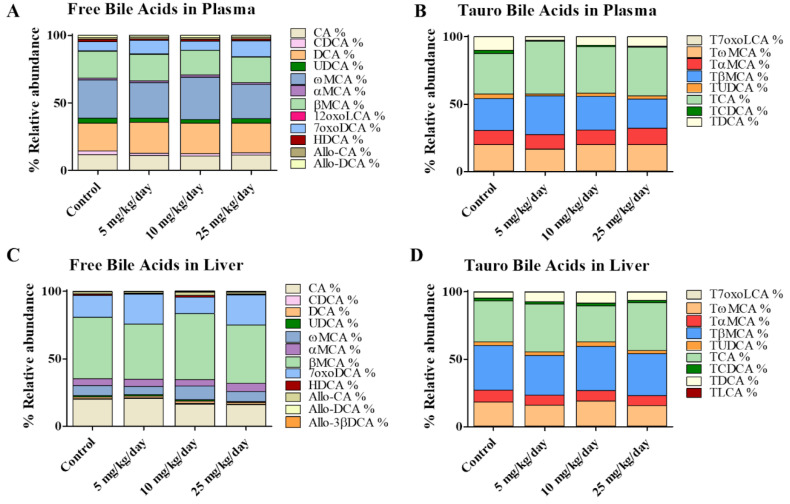
Bile acid composition analyzed in mouse plasma and liver tissue samples. Data represent relative abundances (%). The four groups correspond to control, and simvastatin exposure in mice at three different doses: 5, 10, and 25 mg/kg/day. (**A**,**C**) Profiles of free bile acids in plasma and liver tissue, respectively; (**B**,**D**) Taurine-conjugated bile acids in plasma and liver, respectively. *n* = 6 per group.

**Table 1 metabolites-10-00282-t001:** Method characteristics and validation parameters for the quantification of bile acids in serum and plasma by LC-MS/MS. Abbreviations: CE: collision energy; RT: retention time; R2: linearity; LOD: limit of detection; LOQ: limit of quantitation; LC: low concentration; MC: medium concentration; HC: high concentration, ND: not detected.

Compound	Abbreviation	Transition	ESI	CE (V)	RT (min)	r2	LOD (ng/mL)	LOQ (ng/mL)	Plasma Extraction Recovery			Stability at 4 °C *	
									LC (*n* = 6)	MC (*n* = 6)	HC (*n* = 6)	Plasma Calibration Curve (%)	Plasma Sample (%)
Cholic Acid	CA	373.3 > 355.2	+	8	9.81	0.999	0.45	1.36	111 ± 4	78 ± 6	83 ± 6	2.3	2.7
Chenodeoxycholic acid	CDCA	357.2 > 104.9	+	50	13.76	0.997	0.03	0.09	104 ± 7	81 ± 5	80 ± 6	0.3	0.5
ω-muricholic acid	ωMCA	373.3 > 159.1	+	20	6.29	0.999	0.01	0.02	113 ± 12	77 ± 6	85 ± 7	0.7	0.1
α-muricholic acid	αMCA	373.3 > 355.2	+	15	6.50	0.992	0.17	0.52	98 ± 11	76 ± 5	86 ± 10	1.1	2
β-muricholic acid	βMCA	391.3 > 355.2	+	16	6.85	0.997	0.10	0.30	94 ± 15	75 ± 4	83 ± 6	0.4	1.9
γ-muricholic acid	γMCA	373.3 > 355	+	10	8.14	0.991	0.29	0.86	87 ± 3	79 ± 6	85 ± 6	0.3	ND
Ursodeoxycholic acid	UDCA	357.2 > 95	+	35	9.92	0.997	0.04	0.12	94 ± 11	86 ± 4	86 ± 5	0.5	ND
Hyodeoxycholic acid	HDCA	357.2 > 95.1	+	40	10.34	0.994	0.41	1.23	85 ± 22	84 ± 4	88 ± 5	0.1	3.5
Deoxycholic acid	DCA	391.3 > 345	-	36	14.14	0.991	0.11	0.33	135 ± 9	80 ± 5	84 ± 5	1.2	3.9
Lithocholic acid	LCA	359.3 > 135.1	+	24	17.19	0.997	0.48	1.46	<LOD	66 ± 6	82 ± 5	1.2	ND
3-oxo-lithocholic acid	3oxoLCA	357.3 > 80.9	+	48	17.43	0.992	0.67	2.02	<LOD	82 ± 5	85 ± 7	3	ND
7-oxolithocholic acid	7oxoLCA	373.3 > 355.1	+	35	11.80	0.995	0.82	2.49	75 ± 15	72 ± 4	83 ± 6	1.4	ND
6,7-dioxo-lithocholic acid	6,7dioxoLCA	405.3 > 351.1	+	32	11.71	0.998	0.08	0.24	<LOD	1 ± 0.3	1 ± 0.2	0.7	ND
12-oxo-lithocholic acid	12oxoLCA	391.3 > 145.1	+	12	12.25	0.998	0.68	2.07	112 ± 13	68 ± 4	81 ± 5	2.5	0.4
7-oxo-deoxycholic acid	7oxoDCA	371.3 > 353.2	+	8	6.81	0.990	0.44	1.34	111 ± 15	83 ± 7	85 ± 8	2.3	2.6
Allo-cholic acid	Allo-CA	373.3 > 355.2	+	12	9.44	0.998	0.06	0.19	90 ± 9	77 ± 5	83 ± 6	0.3	1.7
Allo-deoxycholic acid	Allo-DCA	391.3 > 345	-	36	13.91	0.990	0.14	0.41	100 ± 12	64 ± 6	87 ± 6	0.7	5.9
Allo-3β-deoxycholic acid	Allo-3βDCA	391.3 > 345.1	-	40	10.10	0.992	0.51	1.53	86 ± 7	81 ± 5	83 ± 6	0.7	2.9
Allo-12β-deoxycholic acid	Allo-12βDCA	391.3 > 345.1	-	36	11.94	0.994	0.09	0.28	<LOD	80 ± 7	90 ± 6	0.2	ND
Allo-3β-lithocholic acid	Allo-3βLCA	359.3 > 135.1	+	25	15.56	0.987	0.75	2.26	<LOD	54 ± 5	82 ± 4	1.6	ND
Allo-lithocholic acid	Allo-LCA	359.3 > 135.1	+	25	17.08	0.995	0.45	1.37	<LOD	87 ± 4	91 ± 7	3.5	ND
Tauro-cholic acid	TCA	480.3 > 461.9	+	8	5.45	0.999	0.12	0.38	136 ± 15	82 ± 5	83 ± 5	0.3	2.4
Tauro-chenodeoxycholic acid	TCDCA	464.2 > 126	+	28	7.91	0.998	0.05	0.14	109 ± 8	80 ± 5	80 ± 6	0.1	2.4
Tauro-ω-muricholic acid	TωMCA	480.3 > 126	+	24	2.82	0.996	0.06	0.17	128 ± 21	77 ± 5	82 ± 5	0.1	2.7
Tauro-α-muricholic acid	TαMCA	480.3 > 126	+	24	3.07	0.996	0.06	0.17	129 ± 15	79 ± 6	81 ± 5	0.5	2.6
Tauro-β-muricholic acid	TβMCA	480.3> 126	+	24	3.30	0.998	0.04	0.11	131 ± 16	77 ± 4	78 ± 6	0.4	0.8
Tauro-ursodeoxycholic acid	TUDCA	464.2 > 126	+	28	5.20	0.995	0.03	0.10	105 ± 15	80 ± 5	83 ± 6	0.1	0.1
Tauro-deoxycholic acid	TDCA	498.2 > 124.2	-	45	8.79	0.995	0.05	0.14	117 ± 3	80 ± 8	85 ± 3	0.4	0.8
Tauro-7-oxolithocholic acid	T7oxoLCA	480.3 > 126	+	24	5.84	0.996	0.05	0.15	89 ± 5	78 ± 4	83 ± 6	0.5	ND
Tauro-lithocholic acid	TLCA	482.2 > 80	-	56	13.11	0.995	0.08	0.25	107 ± 5	70 ± 5	82 ± 5	0.9	ND
Glyco-cholic acid	GCA	464.4 >74	-	37	6.63	0.998	0.02	0.05	85 ± 5	81 ± 5	84 ± 6	0.4	2.8
Glyco-chenodeoxycholic acid	GCDCA	448.2 > 74	-	30	10.58	0.994	0.02	0.08	98 ± 5	82 ± 5	86 ± 6	0.8	0.5
Glyco-ursodeoxycholic acid	GUDCA	448.2 > 74	-	37	6.43	0.998	0.02	0.06	99 ± 12	80 ± 4	85 ± 6	0.3	3.4
Glyco-deoxycholic acid	GDCA	448.2 > 74	-	30	11.43	0.998	0.011	0.033	97 ± 9	81 ± 4	87 ± 6	0.4	ND
Glyco-lithocholic acid	GLCA	432.2 > 74	-	41	14.59	0.995	1.18	3.58	103 ± 13	70 ± 4	78 ± 5	0.1	ND
Glyco-7-oxo-lithocholic acid	G7oxoLCA	446.2 > 74	-	37	7.63	0.998	0.05	0.16	84 ± 12	79 ± 5	83 ± 6	1.7	ND

* Percentage of variation from the first to the second injection after one week at 4 °C.

**Table 2 metabolites-10-00282-t002:** Method characteristics and validation parameters for the quantification bile acids in liver tissue by LC-MS/MS. Abbreviations: CE: collision energy; RT: retention time; R2: linearity; LOD: limit of detection; LOQ: limit of quantitation; LC: low concentration; MC: medium concentration; HC: high concentration, ND: not detected.

Compound	Abbreviation	Transition	ESI	CE (V)	RT (min)	r2	LOD (ng/mL)	LOQ (ng/mL)	Liver Extraction Recovery			Stability at 4 °C *	
									LC (*n* = 6)	MC (*n* = 6)	HC (*n* = 6)	Liver Calibration Curve (%)	Liver Sample (%)
Cholic Acid	CA	373.3 > 355.2	+	8	9.81	0.999	0.60	1.81	103 ± 3	93 ± 4	104 ± 9	0.3	3.6
Chenodeoxycholic acid	CDCA	357.2 > 104.9	+	50	13.76	0.991	1.46	4.41	<LOD	84 ± 8	97 ± 5	0.5	0.4
ω-muricholic acid	ωMCA	373.3 > 159.1	+	20	6.29	0.998	0.06	0.18	105 ± 6	83 ± 5	101 ± 5	0.8	3.3
α-muricholic acid	αMCA	373.3 > 355.2	+	15	6.50	0.999	0.04	0.13	95 ± 10	87 ± 4	98 ± 3	0.5	2.3
β-muricholic acid	βMCA	391.3 > 355.2	+	16	6.85	0.994	0.32	0.97	101 ± 9	85 ± 5	103 ± 7	0	3.6
γ-muricholic acid	γMCA	373.3 > 355	+	10	8.14	0.997	0.58	1.76	87 ± 6	93 ± 4	96 ± 4	0.2	ND
Ursodeoxycholic acid	UDCA	357.2 > 95	+	35	9.92	0.995	0.10	0.30	<LOD	94 ± 10	105 ± 9	3.1	6.2
Hyodeoxycholic acid	HDCA	357.2 > 95.1	+	40	10.34	0.998	0.58	1.74	<LOD	92 ± 6	95 ± 6	0.5	2.6
Deoxycholic acid	DCA	391.3 > 345	-	36	14.14	0.996	0.64	1.94	88 ± 14	80 ± 6	96 ± 9	0.1	1.5
Lithocholic acid	LCA	359.3 > 135.1	+	24	17.19	0.979	1.20	3.64	<LOD	83 ± 5	105 ± 5	0.4	ND
3-oxo-lithocholic acid	3oxoLCA	357.3 > 80.9	+	48	17.43	0.994	0.70	2.13	<LOD	89 ± 26	104 ± 10	0.5	ND
7-oxolithocholic acid	7oxoLCA	373.3 > 355.1	+	35	11.80	0.998	1.07	3.26	<LOD	96 ± 4	106 ± 11	0.7	ND
12-oxo-lithocholic acid	12oxoLCA	391.3 > 145.1	+	32	12.25	0.999	2.55	7.72	99 ± 10	89 ± 4	99 ± 9	0.9	0.1
6,7-dioxo-lithocholic acid	6,7dioxoLCA	405.3 > 351.1	+	12	11.71	0.999	6.19	18.76	100 ± 10	86 ± 4	103 ± 9	3.1	ND
7-oxo-deoxycholic acid	7oxoDCA	371.3 > 353.2	+	8	6.81	0.995	0.038	0.115	92 ± 6	95 ± 10	116 ± 9	1	1.8
Allo-cholic acid	Allo-CA	373.3 > 355.2	+	12	9.44	0.998	0.14	0.41	88 ± 4	95 ± 4	105 ± 7	1.8	0.7
Allo-deoxycholic acid	Allo-DCA	391.3 > 345	-	36	13.91	0.995	0.41	1.23	93 ± 7	93 ± 5	90 ± 6	0.9	ND
Allo-3β-deoxycholic acid	Allo-3βDCA	391.3 > 345.1	-	40	10.10	0.999	0.36	1.08	92 ± 6	89 ± 4	97 ± 7	0.1	ND
Allo-12β-deoxycholic acid	Allo-12βDCA	391.3 > 345.1	-	36	11.94	0.991	0.03	0.10	<LOD	99 ± 15	97 ± 9	0.6	ND
Allo-3β-lithocholic acid	Allo-3βLCA	359.3 > 135.1	+	25	15.56	0.995	0.45	1.36	<LOD	104 ± 3	98 ± 4	1.3	2.4
Allo-lithocholic acid	Allo-LCA	359.3 > 135.1	+	25	17.08	0.997	0.17	0.52	<LOD	91 ± 10	84 ± 8	2.4	ND
Tauro-cholic acid	TCA	480.3 > 461.9	+	8	5.45	0.998	0.30	0.89	137 ± 10	88 ± 5	96 ± 5	0.5	3.9
Tauro-chenodeoxycholic acid	TCDCA	464.2 > 126	+	28	7.91	0.992	0.52	1.56	103 ± 15	84 ± 5	108 ± 7	0.6	0.3
Tauro-ω-muricholic acid	TωMCA	480.3 > 126	+	24	2.82	0.998	0.149	0.452	143 ± 17	95 ± 6	108 ± 5	0.6	0.9
Tauro-α-muricholic acid	TαMCA	480.3 > 126	+	24	3.07	0.999	0.10	0.30	127 ± 8	94 ± 6	106 ± 4	1	1.9
Tauro-β-muricholic acid	TβMCA	480.3> 126	+	24	3.30	0.999	0.09	0.27	189 ± 29	91 ± 7	108 ± 7	0	0.6
Tauro-ursodeoxycholic acid	TUDCA	464.2 > 126	+	28	5.20	0.993	0.45	1.35	104 ± 10	92 ± 6	101 ± 10	0.2	1.5
Tauro-deoxycholic acid	TDCA	498.2 > 124.2	-	45	8.79	0.991	0.82	2.48	78 ± 10	95 ± 8	107 ± 8	0.8	3.5
Tauro-7-oxolithocholic acid	T7oxoLCA	480.3 > 126	+	24	5.84	0.998	0.18	0.55	98 ± 6	87 ± 6	105 ± 5	0.2	4.5
Tauro-lithocholic acid	TLCA	482.2 > 80	-	56	13.11	0.986	7.21	21.86	99 ± 9	93 ± 6	91 ± 5	0.2	1.2
Glyco-cholic acid	GCA	464.4 >74	-	37	6.63	0.997	0.11	0.34	103 ± 3	90 ± 4	103 ± 3	0.5	1.4
Glyco-chenodeoxycholic acid	GCDCA	448.2 > 74	-	30	10.58	0.996	0.05	0.16	100 ± 2	93 ± 3	99 ± 6	2.4	1
Glyco-ursodeoxycholic acid	GUDCA	448.2 > 74	-	37	6.43	0.999	0.03	0.09	102 ± 6	91 ± 7	103 ± 6	2.7	1.2
Glyco-deoxycholic acid	GDCA	448.2 > 74	-	30	11.43	0.993	0.057	0.172	95 ± 3	89 ± 6	101 ± 4	0.5	ND
Glyco-lithocholic acid	GLCA	432.2 > 74	-	41	14.59	0.992	0.13	0.40	87 ± 5	94 ± 5	93 ± 5	3	ND
Glyco-7-oxo-lithocholic acid	G7oxoLCA	446.2 > 74	-	37	7.63	0.999	0.06	0.19	88 ± 4	91 ± 4	96 ± 3	0.4	ND

* Percentage of variation from the first to the second injection after one week at 4 °C.

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
