# Peer review of "Development and Validation of a Highly Sensitive LC-MS/MS Method for the Analysis of Bile Acids in Serum, Plasma, and Liver Tissue Samples"

_metabolites, 2020, doi:10.3390/metabo10070282_

Round 1

Reviewer 1 Report

The manuscript “Development and validation of a highly sensitive LC-MS/MS method for the analysis of bile acids in serum, plasma and liver tissue samples”, by Cristina Gómez, Simon Stücheli, Denise V. Kratschmar, Jamal Bouitbir and Alex Odermatt, presents results on the development of a LC-MS/MS method for the detection and quantification of 36 bile acids in serum, plasma and liver tissue. An interesting feature of this manuscript is that the authors included a study of statin-induced myotoxicity in mice to test their method. This is a hiding scientific contribution that deserves more visibility; perhaps the authors could at least replace the keyword “statin” by “statin myotoxicity”.

In general, the manuscript is easy to understand and follows a logical sequence. The introduction is clear and the methodology (materials and methods) provides enough information on the experimental work. There is a good level of detail in the description of the methods and procedures as well as the statistical analysis applied in the validation of the method.

In the introduction, the authors presented a detailed explanation of the most important bile acids in humans and mice, as well as rats and guinea pigs. This explanation included aspects referred to the interpretation of the occurrence of specific bile acids in liver tissue (samples obtained from a biopsy). In the same way, the authors indicated that the determination of certain bile acids in body fluids could be useful for diagnostic purposes. I consider that this explanation provides strong support in justifying the research work.

Only one detail that needs correction; on line 279, the unit of temperature must be corrected.

According to the authors, the method reported in this manuscript allows a good separation of the peaks compared to previous works. Considering that 36 compounds are separated, and some of these compounds show important chemical similarities, this is a remarkable achievement. However, the LOQ and LOD are also subject to comparison. Are the values of LOQ and LOD of this new method better than the corresponding values in previous works?

The authors suggest that they “achieved an improvement of the method in terms of simplicity” (line 86). Certainly, the extraction method is simple, and a previous study which was not cited (Tagliacozzi et al. (2003) Clin. Chem. Lab. Med. 41(12), 1633-1641) reported also a simple extraction method with different solvents. I wonder if this new extraction method has advantages over the extraction method reported in 2003. If this the case, it worth highlighting this aspect.

In summary, this is a well-written paper and the experiments were carefully planned. Some minor aspects should be clarified to improve the manuscript. After minor revisions, this paper could be considered for publication in Metabolites.

Author Response

Response to reviewers

We thank the editor and reviewers for their supportive comments. We increased the resolution of the figures and uploaded them again. We addressed the issues raised as follows:

#Reviewer 1

The manuscript “Development and validation of a highly sensitive LC-MS/MS method for the analysis of bile acids in serum, plasma and liver tissue samples”, by Cristina Gómez, Simon Stücheli, Denise V. Kratschmar, Jamal Bouitbir and Alex Odermatt, presents results on the development of a LC-MS/MS method for the detection and quantification of 36 bile acids in serum, plasma and liver tissue. An interesting feature of this manuscript is that the authors included a study of statin-induced myotoxicity in mice to test their method. This is a hiding scientific contribution that deserves more visibility; perhaps the authors could at least replace the keyword “statin” by “statin myotoxicity”.

We would like to thank the reviewer for the comments and suggestions. The keyword has been modified.

In general, the manuscript is easy to understand and follows a logical sequence. The introduction is clear and the methodology (materials and methods) provides enough information on the experimental work. There is a good level of detail in the description of the methods and procedures as well as the statistical analysis applied in the validation of the method.

In the introduction, the authors presented a detailed explanation of the most important bile acids in humans and mice, as well as rats and guinea pigs. This explanation included aspects referred to the interpretation of the occurrence of specific bile acids in liver tissue (samples obtained from a biopsy). In the same way, the authors indicated that the determination of certain bile acids in body fluids could be useful for diagnostic purposes. I consider that this explanation provides strong support in justifying the research work.

Only one detail that needs correction; on line 279, the unit of temperature must be corrected.

The unit of temperature has been corrected (now line 292).

According to the authors, the method reported in this manuscript allows a good separation of the peaks compared to previous works. Considering that 36 compounds are separated, and some of these compounds show important chemical similarities, this is a remarkable achievement. However, the LOQ and LOD are also subject to comparison. Are the values of LOQ and LOD of this new method better than the corresponding values in previous works?

As requested, we compared LOQ and LOD values of our method with data available in the previously reported methods. This shows that we could achieve lower values for most of the bile acid metabolites. We now include a comparison of our method with previously published methods from other investigators in section “2.1. Establishment of the LC-MS/MS method for quantification of bile acids”.

The authors suggest that they “achieved an improvement of the method in terms of simplicity” (line 86). Certainly, the extraction method is simple, and a previous study which was not cited (Tagliacozzi et al. (2003) Clin. Chem. Lab. Med. 41(12), 1633-1641) reported also a simple extraction method with different solvents. I wonder if this new extraction method has advantages over the extraction method reported in 2003. If this the case, it worth highlighting this aspect.

We appreciate the reviewer’s comment. While it is true that some other described methods involved simple extraction methods using similar solvents than the present study, we think we included multiple improvements with respect to previously reported methods. The reference has been included (line 79) and the advantages with respect to this method and other published methods are now discussed in section “2.1. Establishment of the LC-MS/MS method for quantification of bile acids”.

The main advantage of our method, when compared with the cited method and some other reported methods referenced in the manuscript, is the sample volume used. In our current method we use 25 µL while in Tagliacozzi’s work, 250 µL of plasma sample were used. The reduced amount of material required represents a great advantage for studies in which sample volume may be limited such as when using rodent models. The LOD values achieved were similar and/or lower in our case for all of the metabolites. Furthermore, we are able to detect and quantify up to 36 bile acids, in contrast to 14 in previous methods. In our case, we require a lower injection volume and achieve a better baseline chromatographic separation of the compounds.

In summary, this is a well-written paper and the experiments were carefully planned. Some minor aspects should be clarified to improve the manuscript. After minor revisions, this paper could be considered for publication in Metabolites.

Reviewer 2 Report

The paper investigated " Development and validation of a highly sensitive LC-MS/MS method for the analysis of bile acids in serum, plasma and liver tissue samples". The study itself is very important and should be published as soon as possible. Then, many investigators will check (for example, comparison with other methods). Therefore, the methods should be explained in detail enough to be used by other researchers. Authors should add more experimental conditions, for example, Collision Energy for all metabolites.

Author Response

Response to reviewers

We thank the editor and reviewers for their supportive comments. We increased the resolution of the figures and uploaded them again. We addressed the issues raised as follows:

#Reviewer 2

The paper investigated " Development and validation of a highly sensitive LC-MS/MS method for the analysis of bile acids in serum, plasma and liver tissue samples". The study itself is very important and should be published as soon as possible. Then, many investigators will check (for example, comparison with other methods). Therefore, the methods should be explained in detail enough to be used by other researchers. Authors should add more experimental conditions, for example, Collision Energy for all metabolites.

We thank the reviewer for the comments. We have included a comparison of our method with previously reported methods  in section “2.1. Establishment of the LC-MS/MS method for quantification of bile acids”, explaining the advantages of the presented method. Furthermore, the values of collision energy used for each compound have been included in Table 1 and 2.